# COMBINING DIVERSE FEATURE PRIORS

## ABSTRACT

To improve model generalization, model designers often restrict the features that their models use, either implicitly or explicitly. In this work, we explore the design space of leveraging such *feature priors* by viewing them as distinct perspectives on the data. Specifically, we find that models trained with diverse sets of feature priors have less overlapping failure modes, and can thus be combined more effectively. Moreover, we demonstrate that jointly training such models on additional (unlabeled) data allows them to correct each other's mistakes, which, in turn, leads to better generalization and resilience to spurious correlations.

## 1 INTRODUCTION

The driving force behind deep learning's success is its ability to automatically discover predictive features in complex high-dimensional datasets. In fact, these features can generalize beyond the specific task at hand, thus enabling models to transfer to other (yet similar) tasks (Donahue et al., 2014). At the same time, the set of features that the model learns has a large impact on how well it will perform on unseen inputs, especially in the presence of distribution shift (Ponce et al., 2006; Torralba & Efros, 2011; Sagawa et al., 2020) or spurious correlations (Heinze-Deml & Meinshausen, 2017; Beery et al., 2018; Meinshausen, 2018).

Motivated by this, recent work focuses on encouraging specific modes of behavior by preventing the models from relying on certain features. Examples include suppressing texture features (Geirhos et al., 2019; Wang et al., 2019), avoiding $\ell_p$-non-robust features (Tsipras et al., 2019; Engstrom et al., 2019), or utilizing different parts of the frequency spectrum (Yin et al., 2019).

At a high level, these methods can be thought of as ways of imposing a *feature prior* on the learning process, so as to bias the model towards acquiring features that generalize better. This makes the choice of the feature prior to impose a key design decision. The goal of this work is thus to explore the underlying design space of feature priors and, specifically, to understand:

*How can we effectively harness the diversity of feature priors?*

### OUR CONTRIBUTIONS

In this paper, we cast diverse feature priors as different perspectives on the data and study how they can complement each other. In particular, we aim to understand whether training with distinct priors result in models with non-overlapping failure modes and how such models can be combined to improve generalization. This is particularly relevant in settings where the data is unreliable—e.g, when the training data contains a spurious correlation. From this perspective, we focus our study on two priors that arise naturally in the context of image classification, shape and texture, and investigate the following:

**Feature diversity.** We demonstrate that training models with diverse feature priors results in them making mistakes on different parts of the data distribution, even if they perform similarly in terms of overall accuracy. Further, one can harness this diversity to build model ensembles that are more accurate than those based on combining models which have the same feature prior.

**Combining feature priors on unlabeled data.** When learning from unlabeled data, the choice of feature prior can be especially important. For strategies such as self-training, sub-optimal prediction

rules learned from sparse labeled data can be reinforced when pseudo-labeling the unlabeled data. We show that, in such settings, we can leverage the diversity of feature priors to address these issues. By *jointly* training models with different feature priors on the unlabeled data through the framework of *co-training* Blum & Mitchell (1998), we find that the models can correct each other's mistakes to learn prediction rules that generalize better.

**Learning in the presence of spurious correlations.**   Finally, we want to understand whether combining diverse priors during training, as described above, can prevent models from relying on correlations that are spurious, i.e., correlations that do not hold on the actual distribution of interest. To model such scenarios, we consider a setting where a spurious correlation is present in the training data but we also have access to (unlabeled) data where this correlation does not hold. In this setting, we find that co-training models with diverse feature priors can actually steer them away from such correlations and thus enable them to generalize to the underlying distribution.

Overall, our findings highlight the potential of incorporating distinct feature priors into the training process. We believe that further work along this direction will lead us to models that generalize more reliably.

## 2   BACKGROUND: FEATURE PRIORS IN COMPUTER VISION

When learning from structurally complex data, such as images, relying on raw input features alone (e.g., pixels) is not particularly useful. There has thus been a long line of work on extracting input patterns that can be more effective for prediction. While early approaches, such as SIFT (Lowe, 1999) and HOG (Dalal & Triggs, 2005), leveraged hand-crafted features, these have been by now largely replaced by features that are automatically learned in an end-to-end fashion (Krizhevsky, 2009; Ciregan et al., 2012; Krizhevsky et al., 2012).

Nevertheless, even when features are learned, model designers still tune their models to better suit a particular task via changes in the architecture or training methodology. Such modifications can be thought of as imposing *feature priors*, i.e., priors that bias a model towards a particular set of features. One prominent example here are convolutional neural networks, which are biased towards learning a hierarchy of localized features Fukushima (1980); LeCun et al. (1989). Indeed, such a convolutional prior can be quite powerful: it is sufficient to enable many image synthesis tasks *without any training* Ulyanov et al. (2017).

More recently, there has been work exploring the impact of explicitly restricting the set of features utilized by the model. For instance, Geirhos et al. (2019) demonstrate that training models on stylized inputs (and hence suppressing texture information) can improve model robustness to common corruptions. In a similar vein, Wang et al. (2019) penalize the predictive power of local features to learn shape-biased models that generalize better between image styles. A parallel line of work focuses on training models to be robust to small, worst-case input perturbations using, for example, adversarial training Goodfellow et al. (2015); Madry et al. (2018) or randomized smoothing (Lecuyer et al., 2019; Cohen et al., 2019). Such training biases these models away from non-robust features (Tsipras et al., 2019; Ilyas et al., 2019; Engstrom et al., 2019), which tends to result in them being more aligned with human perception (Tsipras et al., 2019; Kaur et al., 2019), more resilient to certain input corruptions (Ford et al., 2019; Kireev et al., 2021), and better suited for transfer to downstream tasks Utrera et al. (2020); Salman et al. (2020).

## 3   FEATURE PRIORS AS DIFFERENT PERSPECTIVES

As we discussed, the choice of feature prior can have a large effect on what features a model relies on and, by extension, on how well it generalizes to unseen inputs. In fact, one can view such priors as distinct perspectives on the data, capturing different information about the input. In this section, we provide evidence to support this view; specifically, we examine a case study on a pair of feature priors that arise naturally in the context of image classification: shape and texture.

## 3.1 TRAINING SHAPE- AND TEXTURE-BIASED MODELS

In order to train shape- and texture-biased models, we either pre-process the model input or modify the model architecture as follows:

**Shape-biased models.** To suppress texture information in the images, we pre-process our inputs by applying an edge detection algorithm. We consider two such canonical algorithms: the *Canny* algorithm Ding & Goshtasby (2001) which produces a binary edge mask, and the *Sobel* algorithm Sobel & Feldman (1968) which provide a softer edge detection, hence retaining some texture information (see Figures 1b and 1c).

**Texture-biased models.** To prevent the model from relying on the global structure of the image, we utilize a variant of the *BagNet* architecture Brendel & Bethge (2019). This architecture deliberately limits the receptive field of the model, thus forcing it to rely on local features (see Figure 1d).

We visualize all of these priors in Figure 1 and provide implementation details in Appendix A.

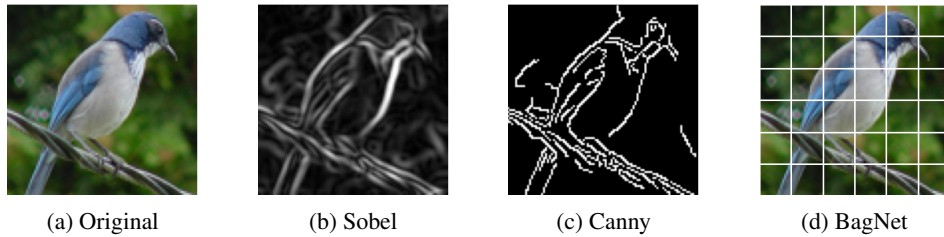

|        (a) Original        |        (b) Sobel        |        (c) Canny        |        (d) BagNet        |

Figure 1: Visualizing different feature priors: (a) an image from the STL-10 dataset; (b) Sobel edge detection; (c) Canny edge detection; (d) the limited receptive field of a BagNet.

|          | CIFAR-10 |       |       |        | STL-10   |       |       |        |
|----------|----------|-------|-------|--------|----------|-------|-------|--------|
|          | Standard | Canny | Sobel | BagNet | Standard | Canny | Sobel | BagNet |
| Standard | 0.598    | 0.237 | 0.259 | 0.38   | 0.554    | 0.305 | 0.385 | 0.357  |
| Canny    |          | 0.545 | 0.324 | **0.143** |          | 0.523 | 0.392 | **0.212** |
| Sobel    |          |       | 0.594 | 0.173  |          |       | 0.649 | 0.262  |
| BagNet   |          |       |       | 0.655  |          |       |       | 0.486  |

Table 2: Correlation (Pearson coefficient) of correct predictions on the test set between different pairs of models. The diagonal entries correspond to models trained with the same prior but from different random initializations. While the two shape-biased models (Sobel and Canny) are more aligned with each other, they are both quite different from the texture-biased model (BagNet).

## 3.2 DIVERSITY OF FEATURE-BIASED MODELS

After training models with shape and texture biases as outlined above, we evaluate whether these models indeed capture complementary information about the input. Specifically, we train models on a small subset (100 examples per class) of the CIFAR-10 (Krizhevsky, 2009) and STL-10 (Coates et al., 2011) datasets, and measure the correlation between which test examples they correctly classify.

We find that pairs consisting of a shape-biased model and a texture-biased model (i.e., Canny and BagNet, or Sobel and BagNet) indeed have the least correlated predictions—cf. Table 2. In other words, the mistakes that these models make are more diverse than those made by identical models trained from different random initializations. At the same time, different shape-biased models (Sobel and Canny) are relatively well-correlated with each other, which corroborates the fact that models trained on similar features of the input are likely to make similar mistakes.

**Model ensembles.** Having shown that training models with these feature priors results in diverse prediction rules, we examine if we can now combine them to improve our generalization. The canonical approach for doing so is to incorporate these models into an ensemble.

We find that the diversity of models trained with different feature priors indeed directly translates into an improved performance when combining them into an ensemble—cf. Table 3. In fact, we find that the performance of the ensemble is tightly connected to prediction similarity of its constituents (as measured in Table 2), i.e., more diverse ensembles tend to perform better. For instance, the best ensemble for the STL-10 dataset is the one combining a shape-biased (Canny) and a texture-biased model (BagNet) which were the models with the least aligned predictions.

|  | Feature Priors | Model 1 | Model 2 | Ensemble |
|---|---|---|---|---|
| Same | Standard + Standard | $52.54 \pm 0.86$ | $51.82 \pm 0.86$ | $54.02 \pm 0.80$ |
|  | Sobel + Sobel | $51.94 \pm 0.84$ | $53.69 \pm 0.82$ | $54.68 \pm 0.83$ |
|  | BagNet + BagNet | $42.22 \pm 0.88$ | $42.56 \pm 0.80$ | $43.49 \pm 0.83$ |
| Different | Standard + Sobel | $52.54 \pm 0.83$ | $51.94 \pm 0.83$ | $\mathbf{58.21 \pm 0.82}$ |
|  | Standard + BagNet | $52.54 \pm 0.84$ | $42.22 \pm 0.84$ | $53.03 \pm 0.81$ |
|  | Sobel + BagNet | $51.94 \pm 0.90$ | $42.22 \pm 0.84$ | $55.14 \pm 0.81$ |

(a) CIFAR-10

|  | Feature Priors | Model 1 | Model 2 | Ensemble |
|---|---|---|---|---|
| Same | Standard + Standard | $53.73 \pm 0.91$ | $55.38 \pm 0.88$ | $57.06 \pm 0.91$ |
|  | Canny + Canny | $56.29 \pm 0.96$ | $54.99 \pm 0.96$ | $58.23 \pm 0.93$ |
|  | BagNet + BagNet | $52.04 \pm 0.98$ | $50.34 \pm 0.94$ | $53.42 \pm 0.93$ |
| Different | Standard + Canny | $53.73 \pm 0.95$ | $56.29 \pm 0.91$ | $\mathbf{60.96 \pm 0.96}$ |
|  | Standard + BagNet | $53.73 \pm 0.98$ | $52.04 \pm 0.90$ | $57.17 \pm 0.90$ |
|  | Canny + BagNet | $56.29 \pm 0.91$ | $52.04 \pm 0.95$ | $\mathbf{61.42 \pm 0.92}$ |

(b) STL-10

Table 3: Ensemble accuracy when combining models trained with a diverse set of feature priors (models with the same prior are trained from different random initialization). Notice how models trained with different priors lead to ensembles with better performance. Moreover, when the accuracy of the two base models is comparable, models that are more diverse (as measured in Table 2) result in better ensembles. We describe the different methods of combining models in Appendix A.4 and provide the full results in Appendix B.2.

## 4 COMBINING DIVERSE PRIORS ON UNLABELED DATA

In the previous section, we saw that training models with different feature priors (e.g., shape- and texture-biased models) can lead to prediction rules with less overlapping failure modes—which, in turn, can lead to more effective model ensembles. However, ensembles only combine model predictions post hoc and thus cannot take advantage of diversity during the training process.

In this section, we instead focus on utilizing diversity *during* training. Specifically, we will leverage the diversity introduced through feature priors in the context of self-training Lee et al. (2013)—a framework commonly used when the labeled data is insufficient to learn a well-generalizing model. This framework utilizes unlabeled data, which are then pseudo-labeled using an existing model and used for further training. While such methods can often improve the overall model performance, they suffer from a significant drawback: models tend to reinforce suboptimal prediction rules even when these rules do not generalize to the underlying distribution Arazo et al. (2020).

Our goal here is thus to leverage diverse feature priors to address this exact shortcoming. Specifically, we will *jointly* train models with different priors on the unlabeled data through the framework of co-training Blum & Mitchell (1998). Since these models capture complementary information about the input (cf. Table 2), we expect them to correct each other's mistakes and improve their prediction rules. As we will see in this section, this approach can indeed have a significant impact

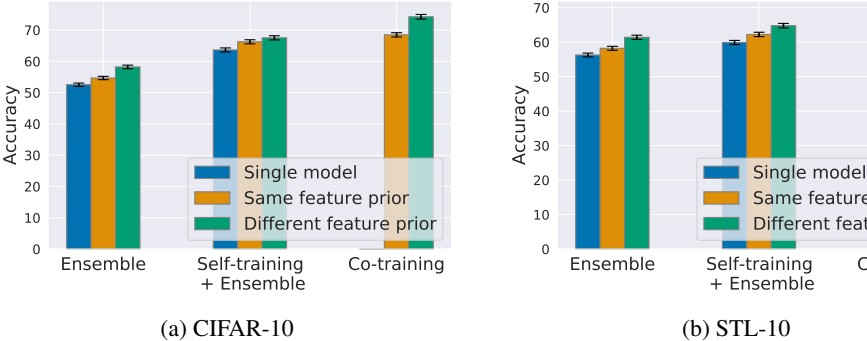

(a) CIFAR-10           (b) STL-10

Figure 4: Test accuracy of pre-trained, self-trained, and co-trained models selecting the best feature prior for each (full results in Table 3, Appendix Table 13, and Table 5 respectively). Notice how combinations of models with different feature priors consistently outperform combinations of models with the same feature prior.

on the performance of the resulting model, outperforming ensembles that combine such models only at evaluation time—see summary in Figure 4.

**Setup.** We base our analysis on the CIFAR-10 and STL-10 datasets. Specifically, we treat a small fraction of the training set as *labeled* examples (100 examples per class), another fraction as our validation set for tuning hyperparameters (10% of the total training examples), and the rest as *unlabeled* data. We report our results on the standard test set of each dataset. (See Appendix A for experimental details, and Appendix B.6 for experiments with varying levels of labeled data.)

### 4.1 SELF-TRAINING AND ENSEMBLES

Before outlining our method for jointly training models with multiple priors, we first describe the standard approach to self-training a single model. At a high level, the predictions of the model on the unlabeled data are treated as correct labels and are then used to further train the same model Lee et al. (2013); Iscen et al. (2019); Zou et al. (2019); Xie et al. (2020). The underlying intuition is that the classifier will predict the correct labels for that data better than chance, and thus these *pseudo-labels* can be used to expand the training set.

In practice, however, these pseudo-labels tend to be noisy. Thus, a common approach is to only use the labels to which the model assigns the highest probability Lee et al. (2013). This process is repeated, self-training on increasingly larger fractions of the unlabeled data until all of it is used. We refer to each such training phase as an *era*.

**Ensembles of diverse self-trained models.** Similarly to our results in Table 3, we find that ensembles comprised of self-trained models with diverse feature priors outperform those that use the same prior from different random initializations (see Figure 4 for a summary and Appendix B.3 for the full results). This demonstrates that, after self-training, these models continue to capture complementary information about the input that can be leveraged to improve performance.

### 4.2 CO-TRAINING MODELS WITH DIFFERENT FEATURE PRIORS

Moving beyond self-training with a single feature prior, our goal in this section is to leverage multiple feature priors by jointly training them on the unlabeled data. This idea naturally fits into the framework of *co-training*: a method used to learn from unlabeled data when inputs correspond to multiple independent sets of features Blum & Mitchell (1998).

Concretely, we first train a model for each feature prior. Then, we collect the pseudo-labels on the unlabeled data that were assigned the highest probability for each model—including duplicates with potentially different labels—to form a new training set which we use for further training. Similarly to the self-training case, we repeat this process over several eras, increasing the fraction of the unlabeled dataset used at each era. Intuitively, this iterative process allows the models to bootstrap

off of each other's predictions, learning correlations that they might fail to learn from the labeled data alone. At the end of this process, we are left with two models, one for each prior, which we combine into a single classifier by training a standard model from scratch on the combined pseudo-labels. We provide a more detailed explanation of the methodology in Appendix A.5.

| Methods | Prior(s) | Labeled Only | +Unlabeled Self/Co-Training | + Standard model with Pseudo-labels |
|---|---|---|---|---|
| Self-training | Standard | $52.54 \pm 0.81$ | $63.65 \pm 0.78$ | $64.02 \pm 0.79$ |
| | Sobel | $51.94 \pm 0.90$ | $63.05 \pm 0.85$ | $64.77 \pm 0.81$ |
| | BagNet | $42.22 \pm 0.81$ | $53.92 \pm 0.84$ | $54.21 \pm 0.81$ |
| Co-training | Standard | $52.54 \pm 0.83$ | $65.06 \pm 0.78$ | $65.10 \pm 0.79$ |
| | +Standard | $51.82 \pm 0.79$ | $64.93 \pm 0.83$ | |
| | Sobel | $51.94 \pm 0.82$ | $\mathbf{71.88 \pm 0.76}$ | $\mathbf{74.25 \pm 0.75}$ |
| | +BagNet | $42.22 \pm 0.80$ | $\mathbf{73.91 \pm 0.73}$ | |

(a) CIFAR-10

| Methods | Prior(s) | Labeled Only | +Unlabeled Self/Co-Training | + Standard model with Pseudo-labels |
|---|---|---|---|---|
| Self-training | Standard | $53.73 \pm 0.94$ | $59.92 \pm 0.93$ | $60.52 \pm 0.91$ |
| | Canny | $56.29 \pm 0.92$ | $58.40 \pm 0.89$ | $62.19 \pm 0.91$ |
| | BagNet | $52.04 \pm 0.92$ | $57.80 \pm 0.99$ | $61.69 \pm 0.96$ |
| Co-training | Standard | $53.73 \pm 0.94$ | $58.05 \pm 0.95$ | $61.16 \pm 0.94$ |
| | +Standard | $55.38 \pm 0.92$ | $60.44 \pm 0.92$ | |
| | Canny | $56.29 \pm 0.94$ | $\mathbf{62.21 \pm 0.93}$ | $\mathbf{67.33 \pm 0.89}$ |
| | +BagNet | $52.04 \pm 1.00$ | $\mathbf{66.74 \pm 0.94}$ | |

(b) STL-10

Table 5: Test accuracy of self-training and co-training methods on STL-10 and CIFAR-10. For each model, we report the original accuracy when trained only labeled data (Column 3) as well as the accuracy after being trained on pseudo-labeled data (Column 4). (Recall that, for the case of co-training, pseudo-labeling is performed by combining the predictions of both models.) Finally, we report the performance of a standard model trained from scratch on the resulting pseudo-labels (Column 5). We provide 95% confidence intervals computed via bootstrap with 5000 iterations.

**Co-training performance.** We find that co-training with shape- and texture-based priors can significantly improve the test accuracy of the final model compared to self-training with any of the priors alone (Table 5). This is despite the fact that, when using self-training alone, the standard model outperforms all other models (Column 4, Table 5). Moreover, co-training models with diverse priors improves upon simply combining them in an ensemble (Appendix B.3).

In Appendix B.5, we report the performance of co-training with every pair of priors. We find that co-training with shape- and texture-based priors together (Canny + BagNet for STL-10 and Sobel + BagNet for CIFAR-10) outperform every other prior combination. Note that this is the case even though, when only ensembling models with different priors (c.f Table 3 and Appendix B.3), Standard + Sobel is consistently the best performing pair for CIFAR-10. Overall, these results indicate that the diversity of shape- and texture-biased models allows them to improve each other over training.

Additionally, we find that, even when training a single model on the pseudo-labels of another model, prior diversity can still help. Specifically, we compare the performance of a standard model trained from scratch using pseudo-labels from various self-trained models (Column 5, Table 5). In this setting, using a self-trained shape- or texture-biased model for pseudo-labeling outperforms using a self-trained standard model. This is despite the fact that, in isolation, the standard model has higher accuracy than the shape- or texture-biased ones (Column 4, Table 5).

**Model alignment over co-training.** To further explore the dynamics of co-training, we evaluate how the correlation between model predictions evolves as the eras progress in Figure 6 (using the prediction alignment measure of Table 2). We find that shape- and texture-biased models exhibit low correlation at the start of co-training, but this correlation increases as co-training progresses. This is in contrast to self-training each model on its own, where the correlation remains relatively low. It is also worth noting that the correlation appears to plateau at a lower value when co-training models with distinct feature priors as opposed to co-training two standard models.

Finally, we find that a standard model trained on the pseudo-labels of other models correlates well with the models themselves (see Appendix B.7). Overall, these findings indicate that models trained on each other's pseudo-labels end up behaving more similarly.

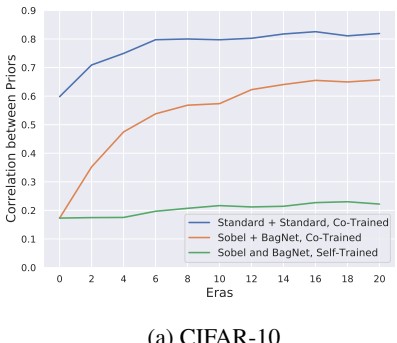 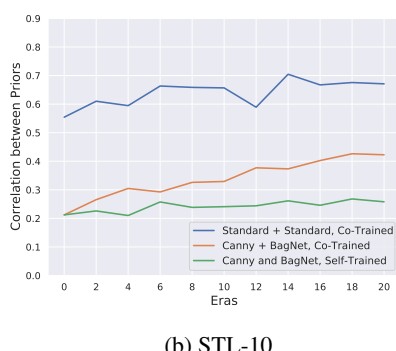

(a) CIFAR-10 (b) STL-10

Figure 6: Correlation between the correct predictions of shape- and texture-biased models over the course of co-training for STL-10 and CIFAR-10. For comparison, we also plot the correlation between the predictions when the models induced by these priors are individually self-trained, as well as the correlation of two standard models when co-trained together.

## 5 Using Co-Training to Avoid Spurious Correlations

A major challenge when training models for real-world deployment is avoiding spurious correlations: associations which are predictive on the training data but not valid for the actual task. Since models are typically trained to maximize train accuracy, they are quite likely to rely on such spurious correlations Gururangan et al. (2018); Beery et al. (2018); Geirhos et al. (2020); Xiao et al. (2020).

In this section, our goal is to leverage diverse feature priors to control the sensitivity of the training process to such spurious correlations. Specifically, we will assume that the spurious correlation does not hold on the unlabeled data (which is likely since unlabeled data can often be collected at a larger scale). Without this assumption, the unlabeled contains no examples that could (potentially) contradict the spurious correlation (we investigate the setting where the unlabeled data is also similarly skewed in Appendix B.10). As we will see, if the problematic correlation is not easily captured by one of the priors, the corresponding model generates pseudo-labels that are inconsistent with this correlation, thus steering other models away from this correlation during co-training.

**Setup.** We study spurious correlations in two settings. First, we create a synthetic dataset by tinting each image of the STL-10 labeled dataset in a class-specific way. This encourages models to rely on the tint, as it is highly predictive on the training set. However, this prediction rule does not generalize to the test set where this correlation is absent. Second, similar to Sagawa et al. (2020), we consider a gender classification task based on CelebA (Liu et al., 2015) where hair color ("blond" vs. "non-blond") is predictive on the labeled data but not on the unlabeled and test data. While gender and hair color are independent attributes on the unlabeled dataset, the labeled dataset consists only of blond females and non-blond males. Similarly to the synthetic case, the labeled data encourages a prediction rule based only on hair color. See Appendix A.1 for details.

**Performance on datasets with spurious features.** We find that, when trained only on the labeled data (where the correlation is fully predictive), both the standard and BagNet models generalize

| Methods | Prior(s) | Labeled Only | +Unlabeled Self/Co-Training | + Standard model with Pseudo-labels |
|---|---|---|---|---|
| Self-training | Standard | $13.99 \pm 0.66$ | $17.56 \pm 0.70$ | $17.81 \pm 0.74$ |
| | Canny | $55.95 \pm 0.92$ | $57.31 \pm 0.89$ | $57.81 \pm 0.92$ |
| | Sobel | $55.11 \pm 0.91$ | $56.12 \pm 0.92$ | $57.16 \pm 0.91$ |
| | BagNet | $13.10 \pm 0.64$ | $13.53 \pm 0.62$ | $14.65 \pm 0.66$ |
| Co-training | Canny | $55.95 \pm 0.90$ | $57.74 \pm 0.90$ | $57.85 \pm 0.95$ |
| | +BagNet | $13.10 \pm 0.65$ | $55.33 \pm 0.92$ | |
| | Sobel | $55.11 \pm 0.95$ | $57.71 \pm 0.90$ | $57.60 \pm 0.94$ |
| | +BagNet | $13.10 \pm 0.62$ | $54.61 \pm 0.94$ | |

(a) Tinted STL-10

| Methods | Prior(s) | Labeled Only | +Unlabeled Self/Co-Training | + Standard model with Pseudo-labels |
|---|---|---|---|---|
| Self-training | Standard | $67.07 \pm 0.58$ | $71.57 \pm 0.53$ | $71.89 \pm 0.53$ |
| | Canny | $80.90 \pm 0.47$ | $85.73 \pm 0.40$ | $86.55 \pm 0.42$ |
| | Sobel | $82.94 \pm 0.45$ | $85.42 \pm 0.43$ | $84.96 \pm 0.43$ |
| | BagNet | $69.35 \pm 0.55$ | $64.89 \pm 0.59$ | $66.15 \pm 0.58$ |
| Co-training | Canny | $80.90 \pm 0.46$ | $89.64 \pm 0.36$ | $91.99 \pm 0.31$ |
| | +BagNet | $69.35 \pm 0.55$ | $91.44 \pm 0.33$ | |
| | Sobel | $82.94 \pm 0.44$ | $90.64 \pm 0.35$ | $90.99 \pm 0.34$ |
| | +BagNet | $69.35 \pm 0.57$ | $88.72 \pm 0.39$ | |

(b) CelebA

Table 7: Test accuracy of self-training and co-training on tinted STL-10 and CelebA, two datasets with spurious features (table structure is identical Table 5). In both datasets, the spurious correlation is more easily captured by the BagNet and Standard models over the shape-based ones. Nevertheless, when co-trained with a shaped-biased model, BagNets are able to significantly improve their performance, indicating that they rely less on this spurious correlation. CI: 95% bootstrap.

poorly in comparison to the shape-biased models (see Table 7). This behavior is expected: the spurious attribute in both datasets is color-related and hence mostly suppressed by the edge detection algorithms used to train shape-based models. Even after self-training on the unlabeled data (where the correlation is absent), the performance of the standard and BagNet models does not improve significantly. Finally, simply ensembling self-trained models post hoc does not improve their performance. Indeed as the texture-biased and standard models are significantly less accurate than the shape-biased one, they end up lowering the overall accuracy of the ensemble (see Appendix B.8).

In contrast, when we co-train a texture-biased model with a shape-biased one, the texture-biased model improves substantially. For instance, when co-trained with a Canny model, the BagNet model improves over self-training by 42% on the tinted STL-10 dataset and 27% on the CelebA dataset. This improvement can be attributed to the fact that the predictions of the shape-biased model are not consistent with the spurious correlation on the unlabeled data. Hence, by being trained on pseudo-labels from that model, the BagNet model is forced to rely on alternative, non-spurious features.

Moreover, particularly on CelebA, the shape-biased model also improves when co-trained with a texture-biased model. This indicates that even though the texture-biased model relies on the spurious correlation, it also captures non-spurious features that, through pseudo-labeling, improve the performance of the shape-based model. In Appendix B.9, we find that these improvements are concentrated on inputs where the spurious correlation does not hold.

## 6 ADDITIONAL RELATED WORK

In Section 2, we discussed the most relevant prior work on implicit or explicit feature priors. Here, we discuss additional related work and how it connects to our approach.

**Shape-biased models.** Several other methods aim to bias models towards shape-based features: input stylization Geirhos et al. (2019); Somavarapu et al. (2020); Li et al. (2021), penalizing early layer predictiveness Wang et al. (2019), jigsaw puzzles Carlucci et al. (2019); Asadi et al. (2019), dropout Shi et al. (2020), or data augmentation Hermann et al. (2020). While, in our work, we choose to suppress texture information via edge detection algorithms, any of these methods can be substituted to generate the shape-based model for our analysis.

**Avoiding spurious correlations.** Other methods that can prevent models from learning spurious correlations include: learning representations that are simultaneously optimal across domains (Arjovsky et al., 2019), enforcing robustness to group shifts (Sagawa et al., 2020), and utilizing multiple data points corresponding to a single physical entity (Heinze-Deml & Meinshausen, 2017). Similar in spirit to our work, these methods aim to learn prediction rules that are supported by multiple views of the data. However, we do not rely on annotations or multiple sources and instead impose feature priors through the model architecture and input preprocessing.

**Pseudo-labeling.** Since the initial proposal of pseudo-labeling for neural networks Lee et al. (2013), there has been a number of more sophisticated pseudo-labeling schemes aimed at improving the accuracy and diversity of the labels Iscen et al. (2019); Augustin & Hein (2020); Xie et al. (2020); Rizve et al. (2021); Huang et al. (2021). In our work, we focus on the simplest scheme for self-labeling—i.e., confidence based example selection. Nevertheless, most of these schemes can be directly incorporated into our framework to potentially improve its overall performance.

A recent line of work explores self-training by analyzing it under different assumptions on the data (Mobahi et al., 2020; Wei et al., 2021; Allen-Zhu & Li, 2020; Kumar et al., 2020). Closest to our work, Chen et al. (2020b) show that self-training on unlabeled data can reduce reliance on spurious correlations under certain assumptions. In contrast, we demonstrate that by leveraging diverse feature priors, we can avoid spurious correlations even if a model heavily relies on them.

**Consistency regularization.** In parallel to pseudo-labeling, consistency regularization is another canonical technique for leveraging unlabeled data. Here, a model is trained to be invariant to a set of input transformations. These transformations might stem from data augmentations and architecture stochasticity Laine & Aila (2017); Berthelot et al. (2019); Chen et al. (2020a); Sohn et al. (2020); Prabhu et al. (2021) or using adversarial examples Miyato et al. (2018).

**Co-training.** One line of work studies co-training from a theoretical perspective (Nigam & Ghani, 2000; Balcan et al., 2005; Goldman & Zhou, 2000). Other work aims to improve co-training by either expanding the settings where it can be applied (Chen et al., 2011) or by improving its stability (Ma et al., 2020; Zhang & Zhou, 2011). Finally, a third line of work applies co-training to images. Since images cannot be separated into disjoint feature sets, one would apply co-training by training multiple models Han et al. (2018), either regularized to be diverse through adversarial examples Qiao et al. (2018) or each trained using a different method Yang et al. (2020). Our method is complementary to these approaches as it relies on explicit feature priors to obtain different views.

## 7 CONCLUSION

In this work, we explored the benefits of combining feature priors with non-overlapping failure modes. By capturing complementary perspectives on the data, models trained with diverse feature priors can offset each others mistakes when combined through methods such as ensembles. Moreover, in the presence of unlabeled data, we can leverage prior diversity by jointly boostrapping models with different priors through co-training. This allows the models to correct each other during training, thus improving pseudo-labeling and controlling for correlations that do not generalize well.

We believe that our work is only the first step in exploring the design space of creating, manipulating, and combining feature priors to improve generalization. In particular, our framework is quite flexible and allows for a number of different design choices, such as choosing other feature priors (cf. Sections 2 and 6), using other methods for pseudo-label selection (e.g., using uncertainty estimation (Lee et al., 2018; Rizve et al., 2021)), and combining pseudo-labels via different ensembling methods. More broadly, we believe that exploring the synthesis of explicit feature priors in new applications is an exciting avenue for further research.

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

# A EXPERIMENTAL DETAILS

## A.1 DATASETS

For our first set of experiments (Section 4), we focus on a canonical setting where a small portion of the training set if labeled and we have access to a pool of unlabeled data.

**STL-10.** The STL-10 Coates et al. (2011) dataset contains 5,000 training and 8,000 test images of size 96×96 from 10 classes. We designate 1,000 of the 5,000 (20%) training examples to be the labeled training set, 500 (10%) to be the validation set, and the rest are used as unlabeled data.

**CIFAR-10.** The CIFAR-10 Krizhevsky (2009) dataset contains 50,000 training and 8,000 test images of size 32×32 from 10 classes. We designate 1,000 of the 50,000 (2%) training examples to be the labeled training set, 5000 (10%) to be the validation set, and the rest as unlabeled data.

In both cases, we report the final performance on the standard test set of that dataset. We also create two datasets that each contain a different spurious correlation.

**Tinted STL-10.** We reuse the STL-10 setup described above, but we add a class-specific tint to each image in the (labeled) training set. Specifically, we hand-pick a different color for each of the 10 classes and then add this color to each of the pixels (ensuring that each RGB channel remains within the valid range)—see Figure 8 for examples. This tint is only present in the labeled part of the training set, the unlabeled and test parts of the dataset are left unaltered.

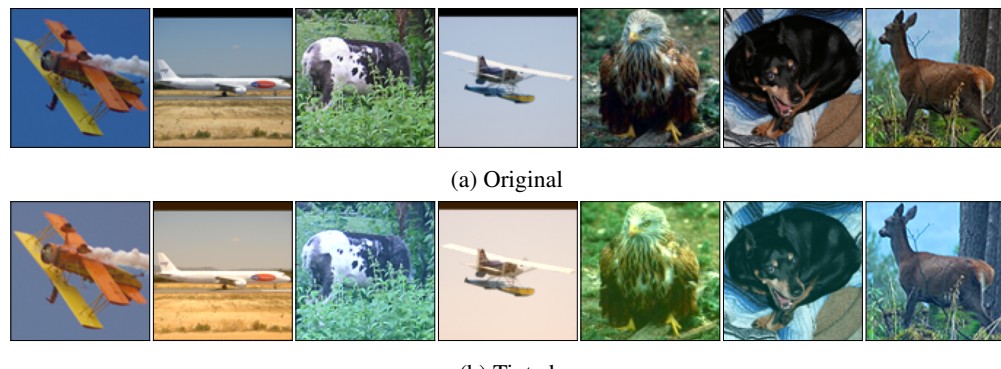

(a) Original

(b) Tinted

Figure 8: Tinted STL-10 images. The tint is class-specific and thus models can learn to predict based mostly on that tint.

**Biased CelebA.** We consider the task of predicting gender in the CelebA Liu et al. (2015) dataset. In order to create a biased training set, we choose a random sample of 500 non-blond males and 500 blond females. We then use a balanced unlabeled dataset consisting of 1,000 random samples for each of: blond males, blond females, non-blond males, and non-blond females. We use the standard CelebA test set which consists of 12.41% blond females, 48.92% non-blond females, 0.90% blond males, and 37.77% non-blond males. (Note that a classifier predicting purely based on hair color with have an accuracy of 50.18% on that test set.)

All of the datasets that we use are freely available for non-commercial research purposes. Moreover, to the best of our knowledge, they do not contain offensive content or identifiable information (other than publicly available celebrity photos).

## A.2 MODEL ARCHITECTURES AND INPUT PREPROCESSING

For both the standard model and the models trained on images processed by edge detection algorithm, we use a standard model architecture—namely, VGG16 Simonyan & Zisserman (2015) with

the addition of batch normalization Ioffe & Szegedy (2015) (often referred to as VGG16-BN). We describe the exact edge detection process as well as the architecture of the BagNet model (texture prior) below. We visualize these priors in Figure 10.

**Canny edge detection.**   Given an image, we first smooth it with a 5 pixel bilateral filter Tomasi & Manduchi (1998), with filter $\sigma$ in the coordinate and color space set to 75. After smoothing, the image is converted to gray-scale. Finally, a Canny filter Canny (1986) is applied to the image, with hysteresis thresholds 100 and 200, to extract the edges.

**Sobel edge detection.**   Given an image, we first upsample it to $128{\times}128$ pixels. Then we convert it to gray-scale and apply a Gaussian blur (kernel size=5, $\sigma = 5$). The image is then passed through a Sobel filter Sobel & Feldman (1968) with a kernel size of 3 in both the horizontal and the vertical direction to extract the image gradients.

**BagNet.**   For our texture-biased model, we use a slimmed down version of the BagNet architecture from Brendel & Bethge (2019). The goal of this architecture is to limit the receptive field of the model, hence forcing it to make predictions based on local features. The exact architecture we used is shown in Figure 9. Intuitively, the top half of the network—i.e., the green and blue blocks— construct features on patches of size $20{\times}20$ for $96{\times}96$ images and $10{\times}10$ for $32{\times}32$ images. The rest of the network consists only of $1{\times}1$ convolutions and max-pooling, hence not utilizing the image's spatial structure.

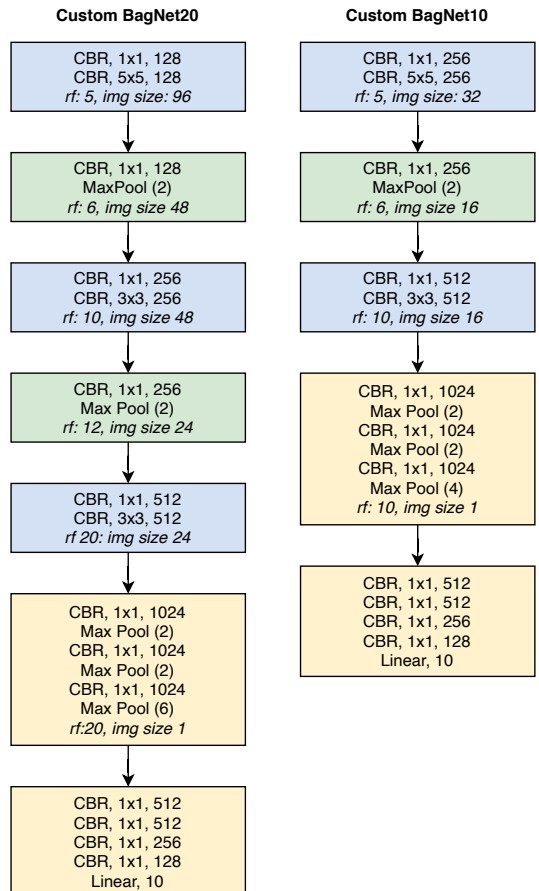

Figure 9: The customized BagNet architecture used for training texture-biased models. The basic building block consists of a convolutional layer, followed by batch normalization and finally a ReLU non-linearity (denoted collectively as *CBR*).

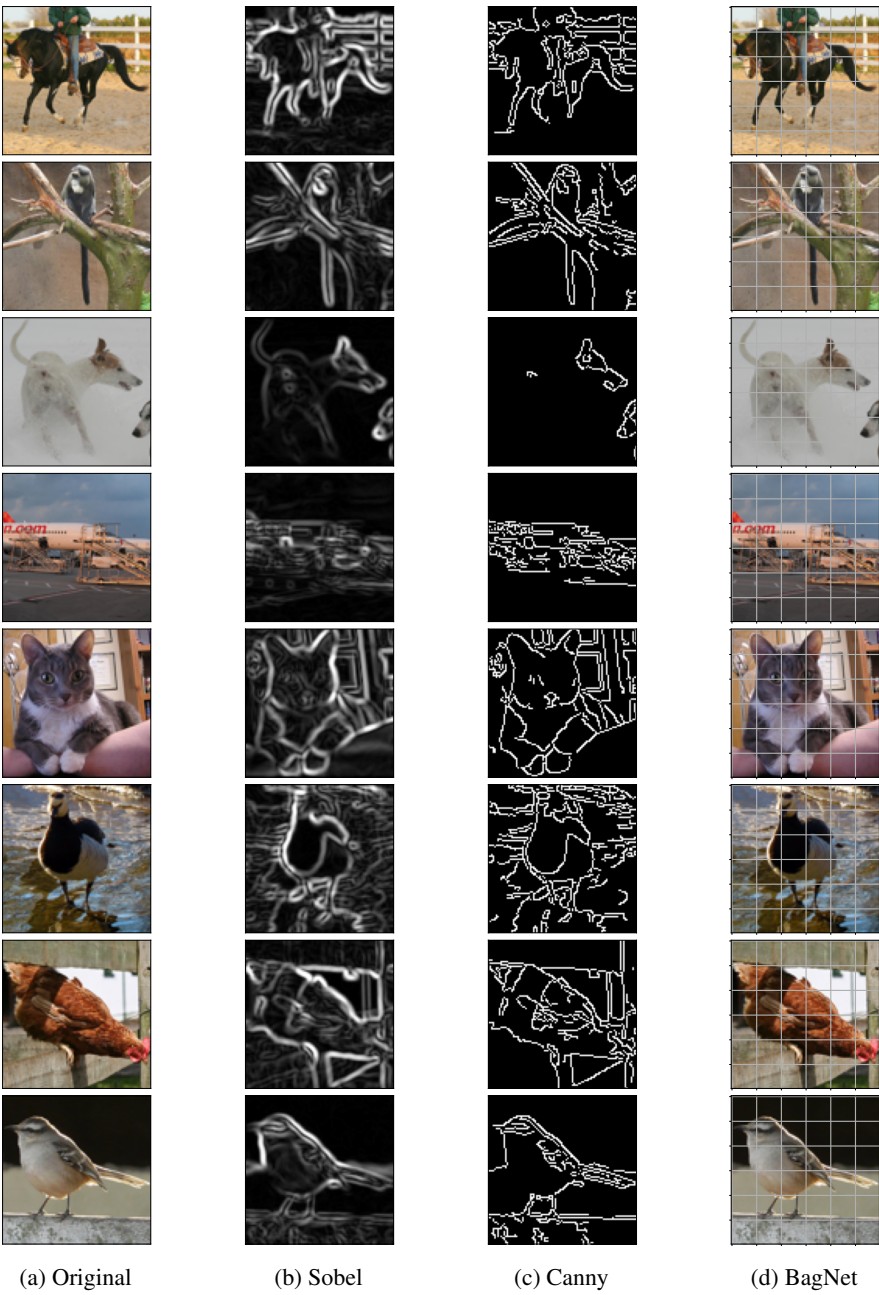

(a) Original      (b) Sobel      (c) Canny      (d) BagNet

Figure 10: Further visualizations of the different feature priors we introduce. For each original image (a), we visualize the output of both edge detection algorithms—Sobel (b) and Canny (c)—as well as the receptive field of the BagNet model.

## A.3 TRAINING SETUP

### A.3.1 BASIC TRAINING

We train all our models using stochastic gradient descent (SGD) with momentum (a coefficient of 0.9) and a decaying learning rate. We add weight decay regularization with a coefficient of $10^{-4}$. In terms of data augmentation, we apply random cropping with a padding of 4 pixels, random horizontal flips, and a random rotation of $\pm 2$ degrees. These transformations are applied after the edge detection processing. We train all models with a batch size of 64 for 96×96-sized images and 128 for 32×32-sized images for a total of 300 epochs. All our experiments are performed using our internal cluster which mainly consists of NVIDIA 1080 Ti GTX GPUs.

**Hyperparameter tuning.** To ensure a fair comparison across feature priors, we selected the hyperparameters for each dataset-prior pair separately, using the held-out validation set (separate from the final test used for reporting performance). Specifically, we performed a grid search choosing the learning rate (LR) from $[0.1, 0.05, 0.02, 0.01, 0.005]$, the number of epochs between each learning rate drop ($K$) from $[50, 100, 300]$ and the factor with which the learning rate is multiplied ($\gamma$) from $[0.5, 1]$. The parameters chosen are shown in Table 11. We found that all models achieved near-optimal performance strictly within the range of each hyperparameters. Thus, we did not consider a wider grid.

| Dataset | Prior | LR | $\gamma$ | $K$ |
|---------|-------|-----|----------|-----|
| STL-10 | Standard | 0.01 | 0.5 | 100 |
| | Canny | 0.01 | 0.5 | 100 |
| | Sobel | 0.005 | 0.5 | 100 |
| | BagNet | 0.05 | 0.5 | 100 |
| CIFAR-10 | Standard | 0.01 | 0.5 | 100 |
| | Canny | 0.01 | 0.5 | 100 |
| | Sobel | 0.01 | 0.5 | 100 |
| | BagNet | 0.01 | 0.1 | 100 |
| CelebA | Standard | 0.005 | 0.5 | 50 |
| | Canny | 0.005 | 0.1 | 100 |
| | Sobel | 0.01 | 0.5 | 50 |
| | BagNet | 0.02 | 0.5 | 100 |

Table 11: Hyperparameters chosen through grid search for each dataset-prior pair (we used the STL-10 hyperparameters for the tinted STL-10 dataset). LR corresponds to the learning rate, $\gamma$ to the factor used to decay the learning rate at each drop, and $K$ to the train epochs between each learning rate drop.

## A.4 ENSEMBLES

In order to leverage prior diversity, we ensemble models trained with (potentially) different priors. We use the following ensembles:

1. **Take Max:** Predict based on the model assigning the highest probability on this example.

2. **Average:** Average the (softmax) output probabilities of the models, predict the class assigned the highest probability.

3. **Rank:** Each model ranks all test examples based on the probability assigned to their predicted labels. Then, for each example, we predict using the model which has a lower rank on this example.

We then report the maximum of these ensemble methods in Table 3.

## A.5 SELF-TRAINING AND CO-TRAINING SCHEMES

In the setting that we are focusing on, we are provided with a labeled dataset $\mathbf{X}$ and an unlabeled dataset $\mathbf{U}$, where typically there is much more unlabeled data ($|\mathbf{U}| \gg |\mathbf{X}|$). We are then choosing a set of (one or more) feature priors each of which corresponds to a different way of training a model (e.g., using edge detection preprocessing).

**General methodology.** We start by training each of these models on the labeled dataset. Then, we combine the predictions of these models to produce pseudo-labels for the unlabeled dataset. Finally, we choose a fraction of the unlabeled data and train the models on that set using the produced pseudo-labels (in additional to the original labeled set $X$). This process is repeated using increasing fractions of the unlabeled dataset until, eventually, models are trained on its entirety. We refer to each such phase as an *era*. We include an additional 5% of the unlabeled data per era, resulting in a total of 20 eras. During each era, we use the training process described in Appendix A.3.1 without re-initializing the models (warm start). After completing this process, we train a standard model from scratch using both the labeled set and resulting pseudo-labels. The methodology used for choosing and combining pseudo-labels is described below for each scheme.

**Self-training.** Since we are only training one model, we only need to decide how to choose the pseudo-labels to use for each era. We do this in the simplest way: at ear $t$, we pick the subset $\mathbf{U_t} \subseteq \mathbf{U}$ of examples that are assigned the highest probability on their predicted label. We attempt to produce a class-balanced training set by applying this process separately on each class (as predicted by the model). The pseudocode for the method is provided in Algorithm 1.

---

**Algorithm 1:** Self-training

---

**Parameters:** Number of eras $T$. Fraction added per era $k$.
**Input** : Labeled data $\mathbf{X}$ with $n$ classes, unlabeled data $\mathbf{U}$, model trained on $\mathbf{X}$.
**for** *era $t \in 1...T$* **do**
    forward-pass $\mathbf{U}$ through the model to create pseudo-labels
    $\mathbf{U_t} = []$
    **for** *each class $c$* **do**
        Select the $\frac{kt|\mathbf{U}|}{n}$ most confident examples from $\mathbf{U}$ predicted by the model as class $c$
        Add those examples to $\mathbf{U_t}$ with class $c$
    Re-train (warm start) the model on $\mathbf{X} \cup \mathbf{U_t}$ until convergence
Train a standard model from scratch on $\mathbf{X} \cup \mathbf{U_T}$.

---

**Standard co-training.** Here, we train multiple models (in our experiments two) based on a common pool of pseudo-labeled examples in each era. In each era $t$, each model labels the unlabeled dataset $\mathbf{U}$. Then, for each class, we alternate between models, adding the next most confident example predicted as that class for that model to $\mathbf{U_t}$, until we reach a fixed number of unique examples have been added for that class (5% of the size of the unlabeled dataset per era). Note that this process allows both conflicts and duplicates: if multiple models are confident about a specific example, that example may be added more than once (potentially with a different label each time). Finally, we train each model (without re-initializing) on $\mathbf{X} \cup \mathbf{U_t}$. The pseudocode for this method can be found in Algorithm 2.

---

**Algorithm 2:** Standard Co-Training

---

**Parameters:** Number of eras $T$. Fraction added per era $k$.
**Input**       : Labeled data $\mathbf{X}$ with $n$ classes, unlabeled data $\mathbf{U}$, models trained on $\mathbf{X}$.
**for** *era $t \in 1...T$* **do**

    forward-pass $\mathbf{U}$ through each model to create pseudo-labels
    $\mathbf{U_t} = []$
    **for** *each class $c$* **do**

        $\mathbf{U_t^{(c)}} = []$
        **while** *the number of unique examples in $\mathbf{U_t^{(c)}} < \frac{kt|\mathbf{U}|}{n}$* **do**

            **for** *each model $m$* **do**

                Add the next most confident example predicted by $m$ as class $c$ to $\mathbf{U_t^{(c)}}$

        Add $\mathbf{U_t^{(c)}}$ to $\mathbf{U_t}$
    Re-train (warm start) each model on $\mathbf{X} \cup \mathbf{U_t}$ until convergence
Train a standard model from scratch on $\mathbf{X} \cup \mathbf{U_T}$.

---

# B ADDITIONAL EXPERIMENTS

## B.1 EXPERIMENT ORGANIZATION

We now provide the full experimental results used to create the plots in the main body as well as additional analysis. Specifically, in Appendix B.2 and B.3 we present the performance of individual ensemble schemes for pre-trained and self-trained models respectively. Then, in Appendix B.5 we present the performance of co-training for each combination of feature priors. In Appendix B.7 we analyse the effect that co-training has on model similarity after training. Finally, in Appendix B.8 we evaluate model ensembles on datasets with spurious correlations and in Appendix B.9 we breakdown the performance of co-training on the skewed CelebA dataset according to different input attributes.

## B.2 FULL PRE-TRAINED ENSEMBLE RESULTS

In Table 3, we reported the best ensemble method for each pair of models trained with different priors on the labeled data. In Table 12, we report the full results over the individual ensembles.

| Feature Priors | Model 1 | Model 2 | Max Conf. | Avg Conf. | Rank | Best |
|---|---|---|---|---|---|---|
| Standard + Standard | $52.54 \pm 0.85$ | $51.82 \pm 0.85$ | $53.98 \pm 0.83$ | $54.02 \pm 0.85$ | $53.98 \pm 0.83$ | $54.02 \pm 0.82$ |
| Sobel + Sobel | $51.94 \pm 0.88$ | $53.69 \pm 0.86$ | $54.62 \pm 0.83$ | $54.68 \pm 0.86$ | $54.61 \pm 0.85$ | $54.68 \pm 0.83$ |
| Canny + Canny | $45.48 \pm 0.84$ | $44.19 \pm 0.88$ | $46.46 \pm 0.82$ | $46.48 \pm 0.86$ | $46.70 \pm 0.83$ | $46.70 \pm 0.79$ |
| BagNet + BagNet | $42.22 \pm 0.80$ | $42.56 \pm 0.83$ | $43.32 \pm 0.82$ | $43.49 \pm 0.82$ | $43.33 \pm 0.85$ | $43.49 \pm 0.84$ |
| Standard + Sobel | $52.54 \pm 0.79$ | $51.94 \pm 0.82$ | $58.14 \pm 0.82$ | $58.21 \pm 0.88$ | $58.12 \pm 0.82$ | $\mathbf{58.21 \pm 0.90}$ |
| Standard + Canny | $52.54 \pm 0.87$ | $45.48 \pm 0.81$ | $55.18 \pm 0.82$ | $55.49 \pm 0.83$ | $54.41 \pm 0.81$ | $55.49 \pm 0.83$ |
| Standard + BagNet | $52.54 \pm 0.85$ | $42.22 \pm 0.80$ | $52.89 \pm 0.84$ | $53.03 \pm 0.89$ | $50.69 \pm 0.81$ | $53.03 \pm 0.85$ |
| Sobel + Canny | $51.94 \pm 0.82$ | $45.48 \pm 0.85$ | $53.81 \pm 0.84$ | $53.95 \pm 0.80$ | $53.18 \pm 0.91$ | $53.95 \pm 0.85$ |
| Sobel + BagNet | $51.94 \pm 0.86$ | $42.22 \pm 0.82$ | $54.42 \pm 0.84$ | $55.14 \pm 0.83$ | $53.50 \pm 0.82$ | $55.14 \pm 0.84$ |
| Canny + BagNet | $45.48 \pm 0.78$ | $42.22 \pm 0.79$ | $49.95 \pm 0.84$ | $50.57 \pm 0.82$ | $49.64 \pm 0.81$ | $50.57 \pm 0.84$ |

(a) Ensemble Baselines for CIFAR-10

| Feature Priors | Model 1 | Model 2 | Max Conf. | Avg Conf. | Rank | Best |
|---|---|---|---|---|---|---|
| Standard + Standard | $53.73 \pm 0.86$ | $55.38 \pm 1.00$ | $56.95 \pm 0.94$ | $57.06 \pm 0.91$ | $56.94 \pm 0.97$ | $57.06 \pm 0.91$ |
| Sobel + Sobel | $55.49 \pm 0.94$ | $55.64 \pm 0.98$ | $56.71 \pm 0.92$ | $56.83 \pm 0.90$ | $56.66 \pm 0.89$ | $56.83 \pm 0.94$ |
| Canny + Canny | $56.29 \pm 0.92$ | $54.99 \pm 0.96$ | $58.04 \pm 0.94$ | $58.23 \pm 0.94$ | $57.95 \pm 0.89$ | $58.23 \pm 0.93$ |
| BagNet + BagNet | $52.04 \pm 0.92$ | $50.34 \pm 0.90$ | $53.40 \pm 0.98$ | $53.42 \pm 0.91$ | $53.29 \pm 0.96$ | $53.42 \pm 0.98$ |
| Standard + Sobel | $53.73 \pm 0.94$ | $55.49 \pm 0.95$ | $59.01 \pm 0.90$ | $59.08 \pm 0.91$ | $58.94 \pm 0.96$ | $59.08 \pm 0.95$ |
| Standard + Canny | $53.73 \pm 1.00$ | $56.29 \pm 0.94$ | $60.90 \pm 0.94$ | $60.96 \pm 0.94$ | $60.85 \pm 0.87$ | $\mathbf{60.96 \pm 0.94}$ |
| Standard + BagNet | $53.73 \pm 0.95$ | $52.04 \pm 0.90$ | $56.99 \pm 0.94$ | $57.17 \pm 0.92$ | $57.04 \pm 0.91$ | $57.17 \pm 0.94$ |
| Sobel + Canny | $55.49 \pm 0.91$ | $56.29 \pm 0.94$ | $59.92 \pm 0.95$ | $60.02 \pm 0.97$ | $59.77 \pm 0.91$ | $60.02 \pm 0.91$ |
| Sobel + BagNet | $55.49 \pm 0.94$ | $52.04 \pm 0.95$ | $59.17 \pm 0.94$ | $59.76 \pm 0.96$ | $59.08 \pm 0.89$ | $59.76 \pm 0.87$ |
| Canny + BagNet | $56.29 \pm 0.96$ | $52.04 \pm 0.95$ | $61.09 \pm 0.92$ | $61.42 \pm 0.94$ | $60.68 \pm 0.92$ | $\mathbf{61.42 \pm 0.93}$ |

(b) Ensemble Baselines for STL-10

Table 12: Full results for ensembles of pre-trained models.

## B.3 ENSEMBLING SELF-TRAINED MODELS

In Table 13, we report the best ensemble method for pairs of self-trained models with different priors. In Table 14, we report the full results over the individual ensembles. We find that, similar to the ensembles of models trained on the labeled data, models with diverse priors gain more from ensembling. However, co-training models with diverse priors together still outperforms ensembling self-trained models.

|  | Feature Priors | Model 1 | Model 2 | Ensemble |
|---|---|---|---|---|
| Same | Standard + Standard | $59.92 \pm 0.95$ | $59.34 \pm 0.88$ | $62.25 \pm 0.93$ |
|  | Canny + Canny | $58.40 \pm 0.94$ | $57.69 \pm 0.94$ | $60.38 \pm 0.92$ |
|  | BagNet + BagNet | $57.80 \pm 0.96$ | $58.11 \pm 0.85$ | $60.52 \pm 0.90$ |
| Different | Standard + Canny | $59.92 \pm 0.90$ | $58.40 \pm 0.95$ | $\mathbf{64.44 \pm 0.90}$ |
|  | Standard + BagNet | $59.92 \pm 0.94$ | $57.80 \pm 0.96$ | $63.19 \pm 0.87$ |
|  | Canny + BagNet | $58.40 \pm 0.94$ | $57.80 \pm 0.96$ | $\mathbf{64.80 \pm 0.91}$ |

(a) STL-10

|  | Feature Priors | Model 1 | Model 2 | Ensemble |
|---|---|---|---|---|
| Same | Standard + Standard | $63.65 \pm 0.81$ | $61.95 \pm 0.82$ | $64.85 \pm 0.79$ |
|  | Sobel + Sobel | $63.05 \pm 0.81$ | $66.01 \pm 0.80$ | $66.25 \pm 0.82$ |
|  | BagNet + BagNet | $53.92 \pm 0.82$ | $52.90 \pm 0.91$ | $55.00 \pm 0.83$ |
| Different | Standard + Sobel | $63.65 \pm 0.81$ | $63.05 \pm 0.83$ | $\mathbf{67.52 \pm 0.77}$ |
|  | Standard + BagNet | $63.65 \pm 0.81$ | $53.92 \pm 0.88$ | $64.10 \pm 0.79$ |
|  | Sobel + BagNet | $63.05 \pm 0.83$ | $53.92 \pm 0.89$ | $65.68 \pm 0.79$ |

(b) CIFAR-10

Table 13: Ensemble performance when combining *self-trained* models with Standard, Canny, Sobel, and BagNet priors. When two models of the same prior are ensembled, the models are trained with different random initializations.

| Feature Priors | Model 1 | Model 2 | Max Conf. | Avg Conf. | Rank | Best |
|---|---|---|---|---|---|---|
| Standard + Standard | $63.65 \pm 0.81$ | $61.95 \pm 0.87$ | $64.84 \pm 0.77$ | $64.85 \pm 0.76$ | $64.83 \pm 0.83$ | $64.85 \pm 0.79$ |
| Sobel + Sobel | $63.05 \pm 0.87$ | $66.01 \pm 0.82$ | $66.19 \pm 0.81$ | $66.25 \pm 0.79$ | $66.17 \pm 0.81$ | $66.25 \pm 0.83$ |
| BagNet + BagNet | $53.92 \pm 0.87$ | $52.90 \pm 0.83$ | $54.86 \pm 0.87$ | $55.00 \pm 0.83$ | $54.87 \pm 0.82$ | $55.00 \pm 0.87$ |
| Standard + Sobel | $63.65 \pm 0.79$ | $63.05 \pm 0.80$ | $67.42 \pm 0.79$ | $67.52 \pm 0.79$ | $67.38 \pm 0.79$ | $\mathbf{67.52 \pm 0.77}$ |
| Standard + Canny | $63.65 \pm 0.90$ | $51.82 \pm 0.88$ | $63.70 \pm 0.81$ | $63.91 \pm 0.81$ | $63.02 \pm 0.83$ | $63.91 \pm 0.82$ |
| Standard + BagNet | $63.65 \pm 0.81$ | $53.92 \pm 0.82$ | $64.05 \pm 0.85$ | $64.10 \pm 0.79$ | $62.69 \pm 0.80$ | $64.10 \pm 0.86$ |
| Sobel + Canny | $63.05 \pm 0.81$ | $51.82 \pm 0.80$ | $61.43 \pm 0.80$ | $61.42 \pm 0.80$ | $60.66 \pm 0.81$ | $61.43 \pm 0.83$ |
| Sobel + BagNet | $63.05 \pm 0.78$ | $53.92 \pm 0.83$ | $65.45 \pm 0.85$ | $65.68 \pm 0.82$ | $64.65 \pm 0.80$ | $65.68 \pm 0.82$ |
| Canny + BagNet | $51.82 \pm 0.81$ | $53.92 \pm 0.79$ | $59.60 \pm 0.81$ | $59.79 \pm 0.83$ | $60.24 \pm 0.82$ | $60.24 \pm 0.81$ |

(a) Ensemble Baselines for CIFAR-10

| Feature Priors | Model 1 | Model 2 | Max Conf. | Avg Conf. | Rank | Best |
|---|---|---|---|---|---|---|
| Standard + Standard | $59.92 \pm 0.92$ | $59.34 \pm 0.99$ | $62.18 \pm 0.92$ | $62.25 \pm 0.96$ | $62.16 \pm 0.88$ | $62.25 \pm 0.94$ |
| Canny + Canny | $58.40 \pm 0.95$ | $57.69 \pm 0.89$ | $60.30 \pm 0.95$ | $60.36 \pm 0.92$ | $60.38 \pm 0.91$ | $60.38 \pm 0.95$ |
| BagNet + BagNet | $57.80 \pm 0.89$ | $58.11 \pm 0.94$ | $60.42 \pm 0.90$ | $60.46 \pm 0.98$ | $60.52 \pm 0.93$ | $60.52 \pm 0.90$ |
| Standard + Sobel | $59.92 \pm 0.92$ | $57.86 \pm 0.91$ | $62.49 \pm 0.89$ | $62.69 \pm 0.91$ | $62.66 \pm 0.89$ | $62.69 \pm 0.94$ |
| Standard + Canny | $59.92 \pm 0.94$ | $58.40 \pm 0.95$ | $64.29 \pm 0.95$ | $64.44 \pm 0.89$ | $64.34 \pm 0.95$ | $\mathbf{64.44 \pm 0.95}$ |
| Standard + BagNet | $59.92 \pm 0.89$ | $57.80 \pm 0.97$ | $63.01 \pm 0.93$ | $63.10 \pm 0.89$ | $63.19 \pm 0.88$ | $63.19 \pm 0.88$ |
| Sobel + Canny | $57.86 \pm 0.91$ | $58.40 \pm 0.93$ | $62.20 \pm 0.92$ | $62.14 \pm 0.92$ | $62.22 \pm 0.90$ | $62.22 \pm 0.91$ |
| Sobel + BagNet | $57.86 \pm 0.95$ | $57.80 \pm 0.95$ | $62.24 \pm 0.94$ | $62.58 \pm 0.90$ | $63.52 \pm 0.91$ | $63.52 \pm 0.88$ |
| Canny + BagNet | $58.40 \pm 0.93$ | $57.80 \pm 0.95$ | $64.38 \pm 0.89$ | $64.64 \pm 0.92$ | $64.80 \pm 0.90$ | $\mathbf{64.80 \pm 0.92}$ |

(b) Ensemble Baselines for STL-10

Table 14: Full results for ensembles of self-trained models.

## B.4 STACKED ENSEMBLING

Here we consider an ensembling technique that leverages a validation set. We implement stacking (also called blending) Töscher et al. (2009); Sill et al. (2009), which takes in the outputs of the member models as input, and then trains a second model to produce the final layer. Here, we take the logits of each model in the ensemble, and train the secondary model using logistic regression on the validation set for the dataset. We report accuracies of the ensemble on the test set below. We again find that prior diversity is important for the performance of the ensemble.

| Feature Priors | Pre-trained | | | Self-trained | | |
|---|---|---|---|---|---|---|
| | Model 1 | Model 2 | Stacked Ensemble | Model 1 | Model 2 | Stacked Ensemble |
| Standard + Standard | $52.54 \pm 0.85$ | $51.82 \pm 0.85$ | $54.13 \pm 0.88$ | $63.65 \pm 0.81$ | $61.95 \pm 0.82$ | $65.13 \pm 0.82$ |
| Sobel + Sobel | $51.94 \pm 0.88$ | $53.69 \pm 0.86$ | $54.46 \pm 0.92$ | $63.05 \pm 0.81$ | $66.01 \pm 0.80$ | $66.35 \pm 0.80$ |
| BagNet + BagNet | $42.22 \pm 0.80$ | $42.56 \pm 0.83$ | $44.28 \pm 0.83$ | $53.92 \pm 0.82$ | $52.90 \pm 0.91$ | $54.94 \pm 0.84$ |
| Standard + Sobel | $52.54 \pm 0.79$ | $51.94 \pm 0.82$ | $\mathbf{57.42 \pm 0.84}$ | $63.65 \pm 0.81$ | $63.05 \pm 0.83$ | $\mathbf{67.01 \pm 0.79}$ |
| Standard + BagNet | $52.54 \pm 0.85$ | $42.22 \pm 0.80$ | $53.65 \pm 0.85$ | $63.65 \pm 0.81$ | $53.92 \pm 0.88$ | $64.61 \pm 0.81$ |
| Sobel + BagNet | $51.94 \pm 0.86$ | $42.22 \pm 0.82$ | $55.75 \pm 0.83$ | $63.05 \pm 0.83$ | $53.92 \pm 0.89$ | $65.67 \pm 0.82$ |

Table 15: Performance of ensembling pre-trained and self-trained models with stacked ensembling on CIFAR-10

| Feature Priors | Pre-trained | | | Self-trained | | |
|---|---|---|---|---|---|---|
| | Model 1 | Model 2 | Stacked Ensemble | Model 1 | Model 2 | Stacked Ensemble |
| Standard + Standard | $53.73 \pm 0.86$ | $55.38 \pm 1.00$ | $56.01 \pm 0.94$ | $59.92 \pm 0.95$ | $59.34 \pm 0.88$ | $60.54 \pm 0.91$ |
| Canny + Canny | $56.29 \pm 0.92$ | $54.99 \pm 0.96$ | $57.70 \pm 0.90$ | $58.40 \pm 0.94$ | $57.69 \pm 0.94$ | $59.23 \pm 0.99$ |
| BagNet + BagNet | $52.04 \pm 0.92$ | $50.34 \pm 0.90$ | $52.35 \pm 0.97$ | $57.80 \pm 0.96$ | $58.11 \pm 0.85$ | $59.48 \pm 0.98$ |
| Standard + Canny | $53.73 \pm 1.00$ | $56.29 \pm 0.94$ | $59.24 \pm 0.88$ | $59.92 \pm 0.90$ | $58.40 \pm 0.95$ | $\mathbf{63.42 \pm 0.89}$ |
| Standard + BagNet | $53.73 \pm 0.95$ | $52.04 \pm 0.90$ | $56.03 \pm 0.98$ | $59.92 \pm 0.94$ | $57.80 \pm 0.96$ | $62.59 \pm 0.91$ |
| Canny + BagNet | $56.29 \pm 0.96$ | $52.04 \pm 0.95$ | $\mathbf{59.98 \pm 0.91}$ | $58.40 \pm 0.94$ | $57.80 \pm 0.96$ | $63.22 \pm 0.94$ |

Table 16: Performance of ensembling pre-trained and self-trained models with stacked ensembling on STL-10

## B.5    SELF-TRAINING AND CO-TRAINING ON STL-10 AND CIFAR-10

| Methods | Prior(s) | Labeled Only | +Unlabeled Self/Co-Training | + Standard model with Pseudo-labels |
|---|---|---|---|---|
| Self-training | Standard | $52.54 \pm 0.86$ | $63.65 \pm 0.76$ | $64.02 \pm 0.82$ |
| | Canny | $45.48 \pm 0.90$ | $51.82 \pm 0.82$ | $55.59 \pm 0.80$ |
| | Sobel | $51.94 \pm 0.88$ | $63.05 \pm 0.84$ | $64.77 \pm 0.80$ |
| | BagNet | $42.22 \pm 0.82$ | $53.92 \pm 0.89$ | $54.21 \pm 0.85$ |
| Co-training | Standard | $52.54 \pm 0.91$ | $65.06 \pm 0.76$ | $65.10 \pm 0.84$ |
| | +Standard | $51.82 \pm 0.86$ | $64.93 \pm 0.80$ | |
| | Canny | $45.48 \pm 0.85$ | $51.15 \pm 0.79$ | $55.74 \pm 0.80$ |
| | +Canny | $44.19 \pm 0.82$ | $51.65 \pm 0.81$ | |
| | Sobel | $51.94 \pm 0.86$ | $67.18 \pm 0.80$ | $68.47 \pm 0.74$ |
| | +Sobel | $53.69 \pm 0.89$ | $67.35 \pm 0.77$ | |
| | Canny | $45.48 \pm 0.79$ | $58.66 \pm 0.81$ | $65.34 \pm 0.81$ |
| | +Sobel | $51.94 \pm 0.80$ | $64.87 \pm 0.79$ | |
| | Canny | $45.48 \pm 0.85$ | $59.19 \pm 0.85$ | $67.59 \pm 0.74$ |
| | +BagNet | $42.22 \pm 0.85$ | $67.92 \pm 0.79$ | |
| | Sobel | $51.94 \pm 0.81$ | $71.88 \pm 0.73$ | $\mathbf{74.25 \pm 0.74}$ |
| | +BagNet | $42.22 \pm 0.82$ | $73.91 \pm 0.71$ | |
| | BagNet | $42.22 \pm 0.79$ | $55.94 \pm 0.83$ | $56.05 \pm 0.77$ |
| | +BagNet | $42.56 \pm 0.86$ | $55.26 \pm 0.88$ | |
| | Canny | $45.48 \pm 0.85$ | $59.23 \pm 0.81$ | $67.21 \pm 0.77$ |
| | +Standard | $52.54 \pm 0.87$ | $66.92 \pm 0.82$ | |
| | Sobel | $51.94 \pm 0.83$ | $71.44 \pm 0.76$ | $\mathbf{73.83 \pm 0.76}$ |
| | +Standard | $52.54 \pm 0.85$ | $73.59 \pm 0.72$ | |
| | Standard | $52.54 \pm 0.88$ | $66.67 \pm 0.83$ | $66.77 \pm 0.75$ |
| | +BagNet | $42.22 \pm 0.80$ | $67.12 \pm 0.75$ | |

Table 17: Performance of self-training and co-training on CIFAR-10 for each prior combination.

| Methods | Prior(s) | Labeled Only | +Unlabeled Self/Co-Training | + Standard model with Pseudo-labels |
|---------|----------|--------------|------------------------------|--------------------------------------|
| Self-training | Standard | $53.73 \pm 0.95$ | $59.92 \pm 0.91$ | $60.52 \pm 0.94$ |
| | Canny | $56.29 \pm 0.96$ | $58.40 \pm 0.91$ | $62.19 \pm 0.92$ |
| | Sobel | $55.49 \pm 0.96$ | $57.86 \pm 0.98$ | $60.92 \pm 0.89$ |
| | BagNet | $52.04 \pm 0.96$ | $57.80 \pm 0.99$ | $61.69 \pm 0.95$ |
| Co-training | Standard | $53.73 \pm 0.95$ | $58.05 \pm 0.92$ | $61.16 \pm 0.95$ |
| | +Standard | $55.38 \pm 0.96$ | $60.44 \pm 0.95$ | |
| | Canny | $56.29 \pm 0.92$ | $60.22 \pm 0.91$ | $63.24 \pm 0.92$ |
| | +Canny | $54.99 \pm 0.94$ | $59.56 \pm 0.94$ | |
| | Sobel | $55.49 \pm 0.96$ | $58.93 \pm 0.91$ | $60.68 \pm 0.94$ |
| | +Sobel | $55.64 \pm 0.95$ | $59.23 \pm 0.90$ | |
| | Canny | $56.29 \pm 0.95$ | $62.40 \pm 0.99$ | $65.53 \pm 0.84$ |
| | +Sobel | $55.49 \pm 0.92$ | $64.11 \pm 0.91$ | |
| | Canny | $56.29 \pm 0.92$ | $62.21 \pm 0.89$ | $\mathbf{67.33 \pm 0.88}$ |
| | +BagNet | $52.04 \pm 0.94$ | $66.74 \pm 0.87$ | |
| | Sobel | $55.49 \pm 0.92$ | $62.72 \pm 0.94$ | $65.79 \pm 0.94$ |
| | +BagNet | $52.04 \pm 1.00$ | $65.44 \pm 0.91$ | |
| | BagNet | $52.04 \pm 0.89$ | $59.85 \pm 0.89$ | $60.84 \pm 0.95$ |
| | +BagNet | $50.34 \pm 0.91$ | $60.16 \pm 0.89$ | |
| | Canny | $56.29 \pm 0.94$ | $62.16 \pm 0.92$ | $65.67 \pm 0.93$ |
| | +Standard | $53.73 \pm 0.92$ | $64.22 \pm 0.91$ | |
| | Sobel | $55.49 \pm 0.95$ | $61.15 \pm 0.89$ | $63.08 \pm 0.91$ |
| | +Standard | $53.73 \pm 0.92$ | $61.74 \pm 0.93$ | |
| | Standard | $53.73 \pm 0.94$ | $61.99 \pm 0.88$ | $62.34 \pm 0.89$ |
| | +BagNet | $52.04 \pm 0.91$ | $62.31 \pm 1.00$ | |

Table 18: Performance of self-training and co-training on STL-10 for each prior combination.

### B.6 CO-TRAINING WITH VARYING AMOUNTS OF LABELED DATA.

In Table 19, we study how the efficacy of combining diverse priors through cotraining changes as the number of labeled examples increase for STL-10. As one might expect, when labeled data is sparse, the feature priors learned by the models alone are relatively brittle: thus, leveraging diverse priors against each other on unlabeled data improves generalization. As the number of labeled examples increases, the models with single feature priors learn more reliable prediction rules that can already generalize, so the additional benefit of combining feature priors diminishes. However, even in settings with plentiful data, combining diverse feature priors can aid generalization if there is a spurious correlation in the labeled data (see Section 5.)

| Number of Labeled Examples | Standard + Standard | Canny + BagNet |
|:---:|:---:|:---:|
| 1000 | $61.16 \pm 0.94$ | $\mathbf{67.33 \pm 0.89}$ |
| 2000 | $68.24 \pm 1.12$ | $\mathbf{72.76 \pm 1.08}$ |
| 3000 | $74.88 \pm 0.97$ | $\mathbf{75.76 \pm 1.04}$ |
| 4000 | $\mathbf{78.85} \pm 0.99$ | $77.44 \pm 1.00$ |

Table 19: Performance of co-training approaches with different amounts of training data for STL-10.

B.7   CORRELATION BETWEEN THE INDIVIDUAL FEATURE-BIASED MODELS AND THE FINAL
STANDARD MODEL

| Method | Prior | CIFAR-10 | | STL-10 | |
|---|---|---|---|---|---|
| | | Before | After | Before | After |
| Self-training | Standard | 0.598 | 0.813 | 0.554 | 0.728 |
| | Canny | 0.237 | 0.622 | 0.305 | 0.519 |
| | Sobel | 0.259 | 0.76 | 0.385 | 0.621 |
| | BagNet | 0.38 | 0.752 | 0.357 | 0.516 |
| Co-training | Canny +BagNet | 0.237 | 0.595 | 0.305 | 0.496 |
| | | 0.38 | 0.664 | 0.357 | 0.538 |
| | Sobel +BagNet | 0.259 | 0.719 | 0.385 | 0.581 |
| | | 0.38 | 0.716 | 0.357 | 0.554 |

Table 20: Similarity between models before and after training on pseudo-labeled data. Our measure of similarity is the (Pearson) correlation between which test examples are correctly predicted by each model. In Columns 3 and 5 we report that notion of similarity between the pre-trained feature-biased models and the pre-trained standard model (the numbers are reproduced from Table 2). Then, in columns 4 and 6 we report the similarity between the feature-biased models at the end of self- or co-training and the standard model trained on their (potentially combined) pseudo-labels. We observe that through this process of training a standard model on the pseudo-labels of different feature-biased models, the former behaves more similar to the latter.

B.8   ENSEMBLES FOR SPURIOUS DATASETS

In Table 21 (full table in Table 22), we ensemble the self-trained priors for the Tinted STL-10 dataset and the CelebA dataset as in Section 5. Both of these datasets have a spurious correlation base on color, which results in a weak Standard and BagNet model. As a result, the ensembles with the Standard or BagNet models do not perform well on the test set. However, in Section 7, we find that co-training in this setting allows the BagNet model to improve when jointly trained with a shape model, thus boosting the final performance.

| Feature Priors | Model 1 | Model 2 | Ensemble |
|---|---|---|---|
| Standard + Canny | $17.56 \pm 0.73$ | $\mathbf{57.31 \pm 0.96}$ | $44.31 \pm 0.90$ |
| Standard + Sobel | $17.56 \pm 0.71$ | $56.12 \pm 0.90$ | $46.06 \pm 0.95$ |
| Standard + BagNet | $17.56 \pm 0.73$ | $13.53 \pm 0.66$ | $16.64 \pm 0.66$ |
| Canny + BagNet | $\mathbf{57.31 \pm 0.96}$ | $13.53 \pm 0.64$ | $48.30 \pm 0.89$ |
| Sobel + BagNet | $56.12 \pm 0.91$ | $13.53 \pm 0.69$ | $49.05 \pm 0.98$ |

(a) Tinted STL-10

| Feature Priors | Model 1 | Model 2 | Ensemble |
|---|---|---|---|
| Standard + Canny | $71.57 \pm 0.53$ | $\mathbf{85.73 \pm 0.40}$ | $84.05 \pm 0.42$ |
| Standard + Sobel | $71.57 \pm 0.55$ | $\mathbf{85.42 \pm 0.43}$ | $82.10 \pm 0.45$ |
| Standard + BagNet | $71.57 \pm 0.53$ | $64.89 \pm 0.56$ | $69.66 \pm 0.55$ |
| Canny + BagNet | $\mathbf{85.73 \pm 0.42}$ | $64.89 \pm 0.56$ | $84.06 \pm 0.45$ |
| Sobel + BagNet | $\mathbf{85.42 \pm 0.43}$ | $64.89 \pm 0.57$ | $82.89 \pm 0.44$ |

(b) CelebA

Table 21: Performance of ensembles consisting of models trained with different priors.

| Feature Priors | Model 1 | Model 2 | Max Conf. | Avg Conf. | Rank | Best |
|---|---|---|---|---|---|---|
| Standard + Canny | 17.56 ± 0.70 | **57.31 ± 0.95** | 44.31 ± 0.98 | 43.48 ± 0.94 | 42.12 ± 0.95 | 44.31 ± 0.94 |
| Standard + Sobel | 17.56 ± 0.66 | 56.12 ± 0.98 | 46.06 ± 0.94 | 44.71 ± 0.91 | 39.39 ± 0.95 | 46.06 ± 0.99 |
| Standard + BagNet | 17.56 ± 0.71 | 13.53 ± 0.64 | 16.59 ± 0.69 | 16.64 ± 0.71 | 16.14 ± 0.74 | 16.64 ± 0.66 |
| Canny + BagNet | **57.31 ± 0.91** | 13.53 ± 0.62 | 48.09 ± 0.96 | 48.30 ± 1.01 | 39.92 ± 0.92 | 48.30 ± 0.95 |
| Sobel + BagNet | 56.12 ± 0.94 | 13.53 ± 0.64 | 49.00 ± 0.95 | 49.05 ± 0.95 | 37.67 ± 0.91 | 49.05 ± 0.93 |

(a) Tinted STL-10

| Feature Priors | Model 1 | Model 2 | Max Conf. | Avg Conf. | Rank | Best |
|---|---|---|---|---|---|---|
| Standard + Canny | 71.57 ± 0.53 | **85.73 ± 0.43** | 83.96 ± 0.44 | 84.05 ± 0.43 | 84.00 ± 0.46 | 84.05 ± 0.43 |
| Standard + Sobel | 71.57 ± 0.57 | **85.42 ± 0.41** | 82.06 ± 0.45 | 82.10 ± 0.45 | 78.01 ± 0.51 | 82.10 ± 0.49 |
| Standard + BagNet | 71.57 ± 0.56 | 64.89 ± 0.56 | 69.66 ± 0.54 | 69.66 ± 0.54 | 68.01 ± 0.58 | 69.66 ± 0.54 |
| Canny + BagNet | **85.73 ± 0.42** | 64.89 ± 0.57 | 84.06 ± 0.44 | 84.06 ± 0.45 | 72.79 ± 0.51 | 84.06 ± 0.44 |
| Sobel + BagNet | **85.42 ± 0.39** | 64.89 ± 0.55 | 82.89 ± 0.46 | 82.89 ± 0.46 | 71.65 ± 0.57 | 82.89 ± 0.43 |

(b) CelebA

Table 22: Performance of individual ensembles on datasets with spurious correlations.

B.9    BREAKDOWN OF TEST ACCURACY FOR CO-TRAINING ON CELEBA

| Method | Prior(s) | Female Blond (N=2480) | Female Not Blond (N=9767) | Male Blond (N=180) | Male Not Blond (N=7535) |
|---|---|---|---|---|---|
| Self-training | Standard | **97.78 $\pm$ 0.52** | 47.06 $\pm$ 0.83 | 55.56 $\pm$ 6.11 | 95.94 $\pm$ 0.37 |
| | Canny | 94.44 $\pm$ 0.81 | 77.27 $\pm$ 0.69 | **78.33 $\pm$ 5.00** | 96.19 $\pm$ 0.36 |
| | Sobel | 95.97 $\pm$ 0.60 | 73.43 $\pm$ 0.78 | 70.56 $\pm$ 5.56 | 96.63 $\pm$ 0.37 |
| | BagNet | **97.26 $\pm$ 0.60** | 35.44 $\pm$ 0.80 | 41.67 $\pm$ 6.67 | 96.30 $\pm$ 0.40 |
| Co-training | Canny +BagNet | 96.94 $\pm$ 0.56 | **86.69 $\pm$ 0.56** | **79.44 $\pm$ 5.00** | 97.53 $\pm$ 0.31 |
| | Sobel +BagNet | 96.81 $\pm$ 0.56 | 84.41 $\pm$ 0.63 | **79.44 $\pm$ 5.00** | **97.89 $\pm$ 0.29** |

Table 23: Accuracy of predicting gender on different subpopulations of the CelebA dataset. We show the accuracy of standard models trained on the pseudo-labels produced by different self- or co-training schemes. Recall that in the training set all females are blond and all males are non-blond (while the unlabeled dataset is balanced). It is thus interesting to consider where this correlation is reversed. We observe that, in these cases, both the standard and BagNet models perform quite poorly, even after being self-trained on the unlabeled dataset where this correlation is absent. At the same time, co-training steers the models away from this correlation, resulting in improved performance. 95% confidence intervals computed via bootstrap are shown.

### B.10 WHAT IF THE UNLABELED DATA ALSO CONTAINED THE SPURIOUS CORRELATION?

In Section 5, we assume that the unlabeled data does not contain the spurious correlation present in the labeled data. This is often the case when unlabeled data can be collected through a more diverse process than labeled data (for example, by scraping the web large scales or by passively collecting data during deployment). This assumption is important: in order to successfully steer models away from the spurious correlation during co-training, the process needs to surface examples which contradict the spurious correlation. However, if the unlabeled data is also heavily skewed, such examples might be rare or non-existent.

What happens if the unlabeled data is as heavily skewed as the labeled data? We return the setting of a spurious association between hair color and gender in CelebA. However, unlike in Section 5, we use an unlabeled dataset that also perfectly correlates hair color and gender – it contains 2000 non-blond males and 2000 blond females. The unlabeled data thus has the same distribution as the labeled data, and contains no examples that reject the spurious correlation (blond males or non-blond females).

| Methods | Prior(s) | Labeled Only | +Unlabeled Self/Co-Training | + Standard model with Pseudo-labels |
|---|---|---|---|---|
| Self-training | Standard | $67.07 \pm 0.57$ | $73.32 \pm 0.55$ | $69.13 \pm 0.58$ |
| | Canny | $80.90 \pm 0.49$ | $80.47 \pm 0.48$ | $76.61 \pm 0.52$ |
| | BagNet | $69.35 \pm 0.55$ | $69.21 \pm 0.53$ | $71.34 \pm 0.54$ |
| Co-training | Canny | $80.90 \pm 0.49$ | $\mathbf{82.17 \pm 0.47}$ | $\mathbf{78.53 \pm 0.49}$ |
| | +BagNet | $69.35 \pm 0.55$ | $76.52 \pm 0.50$ | |

Table 24: Performance of Self-Training and Co-Training techniques when the unlabeled data also contains a complete skew toward hair color (as in the labeled data). 95% confidence intervals computed via bootstrap are shown.

**Self-Training:** Since the unlabeled data follows the spurious correlation between hair color and gender, the standard and BagNet models almost perfectly pseudo-label the unlabeled data. Thus, they are simply increasing the number of examples in the training dataset but maintaining the same overall distribution. Self-training thus does not change the accuracy for models with these priors significantly. In contrast, in the setting in Section 5, there were examples in the unlabeled data which did not align with the spurious correlation (blond males and non-blond females). Since they relied mostly on hair color, the standard and BagNet models actively mislabeled these examples (i.e, by labeling a blond male as female). Training on these erroneous pseudo-labels actively suppressed any features that were not hair color, causing the standard and Bagnet models to perform worse after self-training.

**Co-Training:** In contrast, when performing co-training with the Canny and BagNet priors, the Canny model (which cannot detect hair color) will make mistakes on the unlabeled dataset. These mistakes help are inconsistent with a reliance on hair color: due to this regularization, the BagNet's accuracy improves from 69.35% to 76.52%. Overall, though the gain is not as significant as the setting with a balanced unlabeled dataset, the Canny + BagNet co-trained model can mitigate the pitfalls of the BagNet's reliance on hair color and outperform even the canny self-trained model.

