# OpenReview forum: "Combining Diverse Feature Priors"
_ICLR.cc/2022/Conference — ICLR 2022 Submitted_

### Official Review · Reviewer_uSUu · 2021-10-31

**Correctness:** 2
**Technical Novelty And Significance:** 1
**Empirical Novelty And Significance:** 2
**Recommendation:** 3
**Confidence:** 2

**Main Review:**

Strengths. The experimental part is relatively clear.

Weaknesses. The paper is not very clearly written. First, I would appreciate some (even informal) definition of a feature prior. Later in the text, the co-training is mentioned, and it seems that using different priors = co-training using different views. Is it the idea of the paper? I did not really understand the first contribution: "We demonstrate that training models with diverse feature priors results in them making mistakes on different parts of the data distribution, even if their overall accuracy is similar." What is meant?

As far as I understand, there are two "priors" only explored in the paper: shape and texture.


**Summary Of The Paper:**

The goal of the paper is to improve model generalisation. The authors consider feature priors as distinct perspectives on the data. The results show that models trained with diverse sets of various feature priors have less overlapping modes and are more efficiently combined.

**Summary Of The Review:**

The current contribution is an exploratory work, combining several state-of-the-art methods (for instance, self-training and co-training are used in the experiments).
There is a lack of technical novelty.

---

> ### Author Response · Authors · 2021-11-15
> **Response to Reviewer uSUu**
>
> We thank the reviewer for their time. In the following response, we hope to clarify any terms and points that were unclear.
>
> [**Main goal**] Our overall goal in this work is to show that models with different feature priors (i.e, models that rely on different sets of input features) will make different types of mistakes. They thus can be combined more effectively than two models with the same feature prior -- i.e., that utilize the same set of features -- in order to improve generalization and avoid spurious correlations. We believe that this is a promising, yet relatively unexplored perspective that can be applied to a wide range of areas.
>
> [**Definition of Feature Prior**]. **When we say "feature prior", we refer to a method for biasing the set of features that the model relies on.** For example, a model trained with a texture-based feature prior would primarily rely on texture features, while a model with a shape-based feature prior would primarily rely on shape features. As we describe in Section 2 of the paper, model designers routinely make design decisions that change the features that a model relies on. For example, using convolutions encourages the model to use localized features, and using adversarial training encourages models to rely only on robust features.
>
> In our work, we focus on two natural feature priors (i.e., shape and texture). However, our framework is more general, and could accommodate any two feature priors as long as they are sufficiently diverse. In general, any set of training methodologies that result in models relying on different features of the data can be thought of as diverse feature priors.
>
> [**Clarifying first contribution,** *"training models with diverse feature priors results in them making mistakes on different parts of the data distribution, even if their overall accuracy is similar."*] Here, we train models with different feature priors (shape-based vs. texture-based) and then compare their performance on the test set. Specifically, we investigate whether they make mistakes on a similar set of test examples. We find that models trained with diverse priors make less similar mistakes than models trained with the same prior. We emphasize that this is not simply a consequence of their overall accuracy which is in fact comparable.
>
> (Note that this finding is at the core of our work. The fact that diverse priors lead to different mistakes is what allows us to combine these models into better ensembles as well as improve them during co-training.)

---

> > ### Comment · Reviewer_uSUu · 2021-11-19
> > **Response to authors**
> >
> > I thank the authors for their response. However, I still think that the paper lacks the technical novelty, and the results presented are the results of an empirical study. I keep my initial score.

---

### Official Review · Reviewer_LSTE · 2021-11-01

**Correctness:** 4
**Technical Novelty And Significance:** 3
**Empirical Novelty And Significance:** 3
**Recommendation:** 8
**Confidence:** 4

**Details Of Ethics Concerns:**

No concerns

**Main Review:**

This paper has a number of strengths, that combined makes me recommend the paper for acceptance:
+ The topic of this paper, creating and combining robust, generalizable and diverse feature representations, is of high relevance to a large portion of the ICLR audience.
+ It provides an interesting and valuable formal framework for steering feature representations in different directions, creating multi-view representations of the data.
+ It is well written, well organized, technically correct, and easy to read.
+ The experimental design is sound and well done.

One weakness can be pointed out, not however any cause for not accepting this paper in my opinion:
- The experiments are performed on old datasets, CIFAR-10 and STL-10, both with quite clear class structure and simplistic image setting (e.g. the object centered in the image). It would be interesting to see experiments on more difficult data with fine-grained and hierarchical class structure for example.


**Summary Of The Paper:**

The paper proposes a formalized framework for imposing priors on the feature extraction in deep visual processing models. There has been earlier work on encouraging certain feature representations (e.g. suppressing the focus on texture in feature extraction) and also making feature representations robust to domain shift. The core contribution of this paper is the systematic formulation and investigation of how different, distinct feature priors leads to complementary feature representations that can be combined to provide more robust data representations - in other words, creating synthesized multi-view data representations.
The paper ties back to early (1998) work on co-training (which essentially is multi-modal bootstrapping) and ties this to the more recent body of work on self-supervision and self-training.
Experiments are performed with classical shape- and texture-biased models, and show that the hypothesis - that diverse feature priors are able to robustly create a set of complementary data views - holds.

**Summary Of The Review:**

This paper has a number of strengths, that combined makes me recommend the paper for acceptance: Of high relevance, well written and correct, proposes a valuable framework, and contains sound and well designed experiments.
One weakness can be pointed out, not however any cause for not accepting this paper in my opinion: The experiments are performed on old datasets.

In summary, I propose acceptance for this paper and believe it will be of interest to a large portion of the ICLR audience.

---

> ### Author Response · Authors · 2021-11-15
> **Response to Reviewer LSTE**
>
> We thank the reviewer for their positive review. We do agree that expanding our analysis to settings with more complex class structure (e.g., ImageNet) is a valuable topic for further research.

---

> > ### Comment · Reviewer_LSTE · 2021-11-25
> > **Reply to authors**
> >
> > Thank you for your thorough rebuttal! I am satisfied with your replies to the other reviewers and would like to maintain my rating.

---

### Official Review · Reviewer_KJot · 2021-11-02

**Correctness:** 3
**Technical Novelty And Significance:** 1
**Empirical Novelty And Significance:** 1
**Recommendation:** 3
**Confidence:** 4

**Main Review:**

Positives

+ The study seems to be interesting and maybe useful for practitioners.

Concerns

- Very meagre contribution in terms of technical novelty and framework.

- Looks like an empirical study without much conviction and direction.

- Experimental evaluation and comparisons seem dated, not state of the art.

- The work is very much below the expected standards of ICLR.

**Summary Of The Paper:**

The paper is an empirical study of combining multiple feature priors along with some pre-processing to solve a variety of computer vision tasks.

**Summary Of The Review:**

The paper is clearly below par, can be rejected.

---

> ### Author Response · Authors · 2021-11-15
> **Response to Reviewer KJot**
>
> The goal of our work is to study the potential of incorporating multiple feature priors into the training process. To the best of our knowledge, this rather fundamental question has not been explored in this generality. Our experimental analysis pinpoints when and how these diverse priors can improve model generalization which we believe is of interest to researchers and practitioners alike.
>
> We are not sure if there are any specific points that we could address, but we would be happy to elaborate further.

---

### Official Review · Reviewer_VXFn · 2021-11-02

**Correctness:** 3
**Technical Novelty And Significance:** 2
**Empirical Novelty And Significance:** 3
**Recommendation:** 6
**Confidence:** 4

**Main Review:**

The implementation of shape feature priors (via edge detection preprocessing) and texture feature priors (via limited receptive field via bagnet) makes a lot of sense and does a good job of illustrating a concrete example of a collection of different feature priors.  Some of the co-training experimental results are strong.

One concern I have is that the ensemble results presented in section 3.2 are generated using very primitive ensembling techniques. Appendix A.4 says the combination techniques were simply max, average and lowest rank. It is more common to treat this kind of ensembling as a 2nd-level machine learning problem with the outputs of the models forming the inputs to a 2nd-level model. I would not have expected a fancy 2nd-level model but I was hoping at a minimum that the first-level models would be combined via. e.g. linear regression on first-level outputs from a held-out validation set. Fancier combinations (e.g. neural nets) are also possible, of course. I would encourage the authors to read a few writeups by winners of Kaggle competitions and/or read about the ensembling done in the Netflix Prize to get a better sense of what constitutes state-of-the-art ensemble combination techniques.

I was also concerned about the assumption made on page 7 that spurious correlations are likely not to exist in unlabelled data, because unlabelled data supposedly comes from a more diverse collection process. While this may be true in some cases, there will also be many real-world situations where all the input data, whether labelled or unlabelled, comes from the same distribution and it may all have the same spurious correlation. It is often the case that a small portion of the data is labelled simply because it is very time intensive for humans to do the labeling but nonetheless, the remaining unlabelled data comes from the same distribution. I hope that in future versions of this work, the authors make clear that practitioners should think hard about whether their unlabelled data will have the same spurious correlations as their labelled data, rather than assuming that this is likely the case.

For these reasons, I can only give the paper a 5. I would prefer that the ensemble section be redone with more sophisticated ensembling and/or removed and I would prefer that the absence-of-spurious-correlation-in-unlabelled data assumption be presented more cautiously.

A minor complaint: from a presentation point of view, it is non-standard and a bit strange to add additional related work in section 6. I don't normally expect to read about related work after the results and towards the end of the paper. Related work is normally presented earlier in a paper. The authors might consider moving this section to an earlier point in the paper.

On page 8, I found the bolding of +BagNet cotraining results to be a bit confusing. Normally the 'winning' algorithm results are bolded, which in this case is Canny. I realize that the message of the huge boost of cotraining for +BagNet is what is intended but it still confused me that the bolded numbers were not the best numbers.

It also would be nice to show the method on another domain aside from image classification, although I realize space constraints might make this difficult. The authors might consider removing the ensembling section in future versions of the work and instead using that space for cotraining results on another type of problem.

*** Update after author rebuttal *** In light of the addition of stacking experiments for the ensembling, I have raised my score to a 6.

**Summary Of The Paper:**

The paper presents multiple techniques for training models with different feature priors (i.e. inclinations to focus on different aspects of the training data) and combining them, either post hoc via ensembles or by allowing the models to provide augmented pseudo-labelled training data to each other via co-training. When using simple ensembling techniques, ensembles with a diversity of feature priors are show to perform better than ensembles where the individual models have similar feature priors. Co-training is shown to boost performance substantially when models with diverse feature priors supply pseudo-labels to each other. The problem domain is image classification. The feature priors concern shape and texture. Different preprocessing and or architecture constraint techniques are used for different models so as to predispose them to focus on shape but not texture or vice versa.

**Summary Of The Review:**

The authors achieve some positive results from cotraining of groups image classification models designed to focus on shape but not texture or vice versa. However, their ensembling results are acheived using very primitive ensembling which is not state of the art. They also overstate the odds that spurious correlations are unlikely to exist in unlabelled data.

---

> ### Author Response · Authors · 2021-11-15
> **Response to Reviewer VXFn**
>
> We thank the reviewer for their time and feedback. We address their concerns below.
>
> **[More advanced ensembling techniques]**  Based on our correlation analysis in Table 2, when evaluating models with different priors, there are more examples where at least one of these models is correct (compared to the case where the models are trained identically). Specifically, while these models have comparable accuracy, their errors occur on different parts of the test set. Thus, any reasonable ensemble technique should be able to obtain higher accuracy when combining diverse priors, regardless of the exact method used.
>
> However, we agree that ensembling techniques that use a held out validation set to learn a secondary aggregation model are a useful subset of ensembling techniques. For completeness, we add experiments using a more sophisticated method ([“stacking”](https://arxiv.org/pdf/0911.0460.pdf)) which, using a held out validation set, trains a secondary linear model using logistic regression on the concatenated logits from the member models in order to output a final prediction. We find that the results remain unchanged: diverse ensembles perform better---see new Appendix B.4.
>
> **[Assuming that the unlabeled data does not contain the spurious correlation]** In general, being able to distinguish a spurious from a non-spurious correlation is impossible when observing the training data alone. Thus all methods that aim to learn in the presence of spurious correlations need to make additional assumptions (see corresponding paragraph in Section 6). We decided to focus on our particular setting since there are quite a few realistic scenarios where the unlabeled data is collected through an independent process and thus does not share the same biases as the labeled data.
>
> Nevertheless, we do agree that this is a non-trivial assumption and we thus added additional justification in Section 5 (that we would be happy to further expand upon if needed). Additionally, we added Appendix B.10 where we explore a setting where the spurious correlation is also present in the unlabeled data (such that the labeled and unlabeled data have the same distribution). We find that self-training a model that relies on the spurious correlation (standard or BagNet) only propagates the bias to the unlabeled data---achieving quickly high accuracy---and does not really learn anything new. In contrast, when co-training these models with a shape-biased model, their reliance on the spurious correlation decreases. This can be attributed to the fact that the shape-biased model introduces incorrect labels on certain examples thus clashing with the texture-biased model and biasing it towards the correct features.

---

> > ### Comment · Reviewer_VXFn · 2021-11-22
> > **Thanks for the enhancements**
> >
> > In light of the addition of stacking experiments and also the additional discussion of where spurious correlations are likely to occur, I have raised my score to a 6.

---

### Author Response · Authors · 2021-11-15
**Revisions to Paper**

We thank the reviewers for their feedback. We have uploaded a revised pdf of the paper, which contains the following additions:
- Adding experiments using the "stacking" method of ensembling trained on a held out validation set (Appendix B.4)
- Further justification of the assumptions on the distribution of the unlabeled dataset in the setting of spurious correlations (Section 5)
- An additional experiment investigating co-training with different feature priors when both the labeled and the unlabeled dataset are skewed with the spurious correlation (Appendix B.10)

We additionally respond to reviewer specific feedback in comments below.

Thank you, Authors

---

### Decision · Program_Chairs · 2022-01-20

**Decision:**

Reject

**Comment:**

The manuscript proposes a framework for imposing priors on the feature extraction in deep visual processing models. The core contribution of this manuscript is the systematic formulation and investigation of how different, distinct feature priors leads to complementary feature representations that can be combined to provide more robust data representations. The manuscript uses early work on co-training and also the more recent work on self-supervision and self-training. Experiments are performed with classical shape- and texture-biased models, and show that diverse feature priors are able to robustly create a set of complementary data views.

Positive aspects of the manuscript includes:
1. The topic of this paper, creating and combining robust, generalizable and diverse feature representations, is of high relevance;
2. Positive results from co-training of groups image classification models designed to focus on shape but not texture or vice versa.

There are also several major concerns, including:
1. The ensemble results presented in section 3.2 are generated using very primitive ensembling techniques;
2. The absence-of-spurious-correlation-in-unlabelled data assumption be presented more cautiously;
3. Definition of feature prior;
4. Analysis on another domain aside from image classification.

During the rebuttal period, the Authors provided additional experiments using a more sophisticated method (“stacking”), and additional discussion of where spurious correlations are likely to occur. The manuscript has high rating variance. Some reviewers think that the manuscript lacks the technical novelty, and the results presented are the results of an empirical study. The focus of this manuscript is on two natural feature priors (i.e., shape and texture). It would strengthen the manuscript if the Authors can provide further analysis to emphasise the generality of the proposed framework that it could accommodate any two feature priors as long as they are sufficiently diverse.